# Microbiological Epidemiology of Invasive Infections Due to Non-Beta-Hemolytic Streptococci, France, 2021

Céline Plainvert,[a,b] Erika Matuschek,[c] Nicolas Dmytruk,[a,b] Marine Gaillard,[a,b] Amandine Frigo,[a,b] Yassine Ballaa,[a] Eddy Biesaga,[a] Gunnar Kahlmeter,[c,d] Claire Poyart,[a,b,e] ![ORCID] Asmaa Tazi[a,b,e]

[a]Assistance Publique - Hôpitaux de Paris, Hôpitaux Universitaires Paris Centre Site Cochin, Service de Bactériologie, Paris, France
[b]Centre National de Référence des Streptocoques, Paris, France
[c]EUCAST Development Laboratory, Växjö, Sweden
[d]Department of Clinical Microbiology, Central Hospital, Växjö, Sweden
[e]Université Paris Cité, Paris, France

**ABSTRACT** Non-beta-hemolytic streptococci (NBHS), also referred to as *viridans* streptococci, represent an underestimated cause of human invasive diseases. Their resistance to antibiotics, including beta-lactam agents, often complicate their therapeutic management. A prospective multicenter study was conducted by the French National Reference Center for Streptococci between March and April 2021 to describe the clinical and microbiological epidemiology of invasive infections due to NBHS, excluding pneumococcus. A total of 522 NBHS invasive cases were collected. Distribution among streptococcal groups was: *Streptococcus anginosus* (33%), *Streptococcus mitis* (28%), *Streptococcus sanguinis* (16%), *Streptococcus bovis/equinus* (15%), *Streptococcus salivarius* (8%), and *Streptococcus mutans* (<1%). Median age of infection was 68 years old (range <1 day to 100 years). Cases were more frequent in male patients (gender ratio M/F 2.1:1) and manifested mainly as bacteremia without focus (46%), intra-abdominal infections (18%) and endocarditis (11%). All isolates were susceptible to glycopeptides and displayed low-level inherent gentamicin resistance. All isolates of the *S. bovis/equinus*, *S. anginosus*, and *S. mutans* groups were susceptible to beta-lactams. Conversely, nonsusceptibility to beta-lactams was found in 31%, 28%, and 52% of *S. mitis*, *S. salivarius*, and *S. sanguinis* isolates, respectively. The screening for beta-lactam resistance using the recommended one unit benzylpenicillin disk screening failed to detect 21% of resistant isolates (21/99). Last, overall resistance rates to the alternative anti-streptococcal molecules clindamycin and moxifloxacin were 29% (149/522) and 1.6% (8/505), respectively.

**IMPORTANCE** NBHS are recognized as opportunistic pathogens particularly involved in infections of the elderly and immunocompromised patients. This study underlines their importance as common causes of severe and difficult-to-treat infections such as endocarditis. Although species of the *S. anginosus* and *S. bovis/equinus* groups remain constantly susceptible to beta-lams, resistance in oral streptococci exceeds 30% and screening techniques are not fully reliable. Therefore, accurate species identification and antimicrobial susceptibility testing by MICs determination appears essential for the treatment of NBHS invasive infections, together with continued epidemiological surveillance.

**KEYWORDS** non-beta-hemolytic streptococci, *viridans* streptococci, streptococcal infections, antibiotic susceptibility, beta-lactam resistance, epidemiology

Non-beta-hemolytic streptococci (NBHS), sometimes referred to as *viridans* group streptococci, constitute a heterogeneous group of bacteria belonging to the oropharyngeal, urogenital, and gastrointestinal microbiota. They are currently classified into six major groups, *Streptococcus anginosus*, *Streptococcus bovis/equinus*, *Streptococcus mitis*, *Streptococcus mutans*, *Streptococcus salivarius*, and *Streptococcus sanguinis* groups,

Address correspondence to Asmaa Tazi, asmaa.tazi@aphp.fr.

The authors declare no conflict of interest.

and can be responsible for a wide range of diseases (1, 2). *Streptococcus mutans* and *Streptococcus sobrinus* are commonly associated with dental caries and odontogenic infections (3). Besides, NBHS are a frequent cause of bacteremia in immunocompromised patients resulting from underlying malignancy or cancer therapy, ranging from the first to the fifth cause of bloodstream infections in neutropenic patients with hematological malignancies and solid tumors, and accounting for 8% to 26% of the cases (4–6). NBHS, mainly of the *S. bovis/equinus*, *S. mitis*, *S. salivarius*, and *S. sanguinis* groups, are also reported as the first or second cause of infective endocarditis worldwide, accounting for approximately 20% of the cases (3, 7–9). Additionally, isolates of the *S. anginosus* group are causative agents of bacteremia, pyogenic infections, and abscesses in immunocompetent adults and children (10). Similar to group A and group B beta-hemolytic streptococci (*Streptococcus pyogenes* and *Streptococcus agalactiae*, respectively), an increasing incidence of *S. anginosus* group bacteremia has been reported over the past decades, reaching 3.7 cases per 100,000 inhabitants in 2017 in Canada (11). Infections occur in the absence of any underlying condition or comorbidity in approximately 50% of the cases, are usually polymicrobial mainly in association with obligate anaerobes, and involve various sites, including the abdomen, lower and upper respiratory tract, and the brain (12, 13). Among the three species which constitute this group (*S. anginosus*, *Streptococcus intermedius*, and *Streptococcus constellatus*), *S. anginosus* is the most commonly isolated in both bloodstream infections and abscesses (50% to 70% of the cases) (12–14).

Beta-lactam antibiotics remain the first-line antibiotics for NBHS invasive infections (15). Nonetheless, penicillin-resistant isolates were described as early as the 1960s (16, 17) and increasing rates of resistance are reported since the 1980s (2, 18, 19). As in *Streptococcus pneumoniae*, the loss of susceptibility results from mutations in the genes encoding the penicillin-binding proteins which alter their affinity for one or more beta-lactam molecules. Resistance rates greatly vary between streptococcal groups with penicillin nonsusceptible isolates reaching rates as high as 35% in the *S. mitis* and *S. sanguinis* groups while remaining extremely rare in the *S. anginosus* group (19–21). Thus, resistance to beta-lactam antibiotics is very common in certain streptococcal species and frequently complexifies the therapeutic management of difficult-to-treat infections such as bone and joint infections or infective endocarditis.

Several antibiotics can be used as alternative or adjunctive therapy to beta-lactams in case of allergy, resistance, or in difficult-to-treat infections. These antibiotics include macrolides, lincosamides, and streptogramins (MLS), rifampicin, linezolid, and glycopeptides. Whereas NBHS resistance to MLS, together with that to beta-lactams, is an evolving problem, linezolid and glycopeptides remain usually active (19, 22).

Altogether, the epidemiology of NBHS invasive infections is most often studied in specific clinical settings or by considering a single species or group of species. Besides, large studies of NBHS susceptibility to antibiotics appear relatively scarce. Here, we report at the species level the epidemiology of more than 500 NBHS invasive infections, with the exclusion of pneumococcal infections. We describe for each NBHS group the clinical manifestations and the susceptibility to antibiotics, including five beta-lactam agents, MLS, and glycopeptides using disk diffusion (DD), gradient tests, and broth microdilution (BMD).

## RESULTS

**Microbiological epidemiology of NBHS invasive infections.** Invasive isolates of NBHS were prospectively referred to the French National Reference Centre for Streptococci (Strep-NRC, https://cnr-strep.fr) by correspondents located throughout France. A total of 522 invasive NBHS was reported between March and April 2021. All isolates were identified by mass spectrometry except 27 (5.2%) which required *sodA* sequencing (23). Eventually, 22 different species were identified, each of them consisting of one to 97 isolates (Table S1). The 10 most prevalent species accounted for 85% of cases and included, in the *S. anginosus* group, *S. anginosus* (97; 18.6%), *S. constellatus* (53; 10.2%), and *S. intermedius* (22; 4.2%); in the *S. mitis* group, *Streptococcus oralis* (83; 15.9%), and *S. mitis* (38; 7.3%); in the *S. sanguinis* group, *S. sanguinis* (36; 6.9%), and *Streptococcus parasanguinis* (29; 5.6%); in the

**TABLE 1** Clinical characteristics of the 522 non-beta-hemolytic streptococci invasive infections

| *Streptococcus* group | anginosus | bovis/equinus | mitis | mutans | salivarius | sanguinis | Total | P value |
|---|---|---|---|---|---|---|---|---|
| Sex ratio M/F | 1.9:1 | 2.2:1 | 1.7:1 | 4:1 | 3.3:1 | 2.4:1 | 2.1:1 | ND |
| Age, median | 64 y | 82 y | 65 y | 73 y | 66 y | 71 y | 68 y | ND |
| (min to max) | (2 m to 95 y) | (1 d[h] to 96 y) | (3 m to 100 y) | (26 y to 86 y) | (<1 d to 90 y) | (5 d to 94 y) | (<1 d to 100 y) | |
| Age, interquartile range, in years | 51 to 73 | 69 to 85 | 51 to 76 | 26 to 76 | 55 to 72 | 57 to 81 | 55 to 79 | ND |
| Total, *n* (%) | 172 (33.0%) | 77 (14.8%) | 148 (28.4%) | 5 (1.0%) | 39 (7.5%) | 81 (15.5%) | 522 (100%) | 0.0023[f] |
| | | | | | | | | |
| Adult cases (≥18 y), *n* (%)[a] | 165 (95.9%) | 74 (96.1%) | 135 (91.2%) | 5 (100%) | 34 (87.2%) | 77 (95.1%) | 490 (93.9%) | <0.0001[g] |
| Bacteremia without focus | 57 (33.1%) | 42 (54.5%) | 68 (45.9%) | 3 (60%) | 16 (41.0%) | 35 (43.2%) | 221 (42.3%) | 0.011 |
| Intra-abdominal infections | 41 (23.8%) | 11 (14.3%) | 16 (10.8%) | 0 | 10 (25.6%) | 15 (18.5%) | 93 (17.8%) | 0.024 |
| Endocarditis | 3 (1.7%) | 12 (15.6%) | 18 (12.2%) | 2 (40%) | 2 (5.1%) | 19 (23.5%) | 56 (10.7%) | <0.0001 |
| Pulmonary infections | 26 (15.1%) | 1 (1.3%) | 14 (9.5%) | 0 | 3 (7.7%) | 5 (6.2%) | 49 (9.4%) | 0.010 |
| Bone and joint infections | 13 (7.6%) | 3 (3.9%) | 7 (4.7%) | 0 | 0 | 3 (3.7%) | 26 (5.0%) | ND |
| Skin and soft tissue infections | 14 (8.1%) | 2 (2.6%) | 3 (2.0%) | 0 | 1 (2.6%) | 0 | 20 (3.8%) | ND |
| Urinary tract infections[b] | 4 (2.3%) | 1 (1.3%) | 3 (2.0%) | 0 | 0 | 0 | 8 (1.5%) | ND |
| Upper respiratory tract infections[b] | 4 (2.3%) | 0 | 4 (2.7%) | 0 | 0 | 0 | 8 (1.5%) | ND |
| Central nervous system infections | 1 (0.6%) | 1 (1.3%) | 0 | 0 | 2 (5.1%) | 0 | 4 (0.8%) | ND |
| Others[c] | 2 (1.2%) | 1 (1.3%) | 2 (1.4%) | 0 | 0 | 0 | 5 (1.0%) | ND |
| | | | | | | | | |
| Neonates and infants (birth to ≤1 y), *n* (%)[a] | 1 (0.6%) | 3 (3.9%) | 2 (1.4%) | 0 | 4 (10.3%) | 1 (1.2%) | 11 (2.1%) | ND |
| Early Onset Sepsis (0 to 3 d) | 0 | 2 (2.6%) | 0 | 0 | 2 (5.1%) | 1 (1.2%) | 5 (1.0%) | ND |
| Late Onset Sepsis (4 to 89 d) | 0 | 0 | 0 | 0 | 2 (5.1%) | 0 | 2 (0.4%) | ND |
| Others[d] | 1 (0.6%) | 1 (1.3%) | 2 (1.4%) | 0 | 0 | 0 | 4 (0.8%) | ND |
| | | | | | | | | |
| Pediatric cases (1 to 18 y), *n* (%)[a] | 6 (3.5%) | 0 | 11 (7.4%) | 0 | 1 (2.6%) | 3 (3.7%) | 21 (4.0%) | ND |
| Bacteremia without focus | 1 (0.6%) | 0 | 9 (6.1%) | 0 | 1 (2.6%) | 2 (2.5%) | 13 (2.5%) | ND |
| Others[e] | 5 (2.9%) | 0 | 2 (1.4%) | 0 | 0 | 1 (1.2%) | 8 (1.5%) | ND |

[a]Percentages are relative to the total isolates within each non-beta-hemolytic *Streptococcus* group.
[b]Bacteremia-associated infections.
[c]Including vascular- and catheter-related bloodstream infection (*n* = 3), endophthalmitis (*n* = 1), and upper-genital infection (*n* = 1).
[d]Including pyelonephritis (*n* = 2), bacteremia without focus (*n* = 1), pulmonary infections (*n* = 1).
[e]Including intra-abdominal infections (*n* = 2), endocarditis (*n* = 2), upper respiratory tract infections (*n* = 2), pulmonary infections (*n* = 1), and central nervous system infection (*n* = 1).
[f]P-value for the distribution among adult, neonates and infants, and pediatric cases.
[g]P-value for the distribution of clinical manifestations among adult cases.
[h]d, days; m, months; y, years; ND, not determined.

*S. bovis/equinus* group, *Streptococcus gallolyticus* (30; 5.7%), and *Streptococcus pasteurianus* (24; 4.6%); and in the *S. salivarius* group, *S. salivarius* (30; 5.7%).

The overall gender ratio M/F was of 2.1:1 and the median age was 68 years old (Table 1). Thirty-two cases (6.1%) were pediatric. Bacteremia without focus (*n* = 241, 46.2%), intra-abdominal infections (*n* = 95; 18.2%), endocarditis (*n* = 58; 11.1%), pulmonary infections (*n* = 51; 9.2%), and bone and joint infections (*n* = 26; 5.0%) accounted for 90% of infections. The clinical manifestations varied significantly according to NBHS groups (Fig. 1; Table 1), isolates from the *S. anginosus* group being overrepresented in

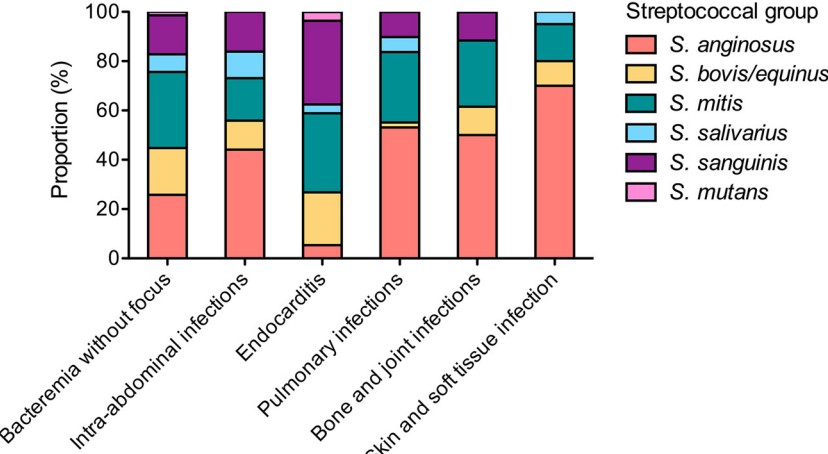

**FIG 1** Distribution of NBHS groups according to the clinical manifestations of adult invasive infections. The proportion of the different NBHS groups according to the main clinical manifestations of invasive infections in adults are represented. Total number of cases: bacteremia without focus, *n* = 221; intra-abdominal infections, *n* = 93; endocarditis, *n* = 56; pulmonary infections, *n* = 49, bone and joint infections, *n* = 26; skin and soft tissue infections, *n* = 20.

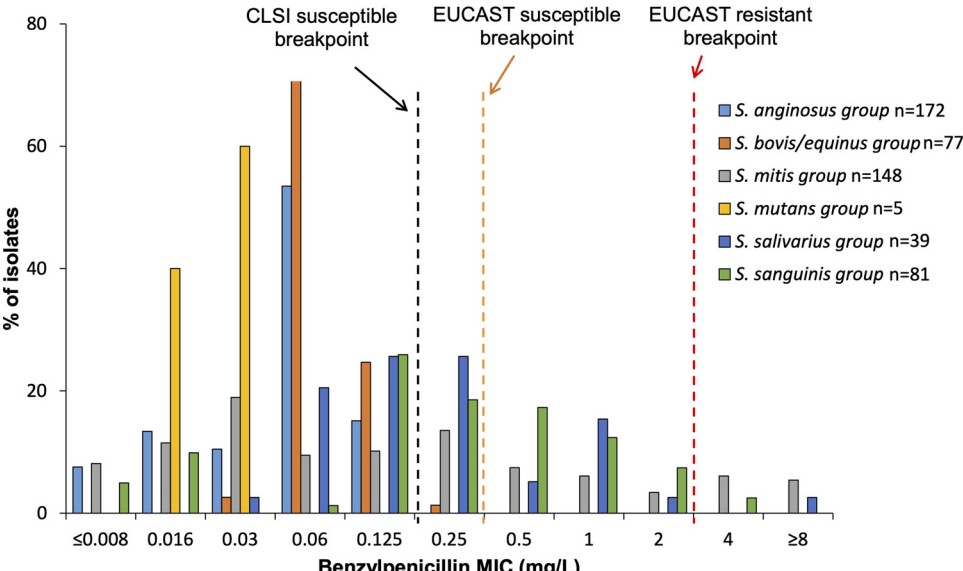

**FIG 2** Distribution of PEN MIC values within each NBHS group (*n* = 522). MIC values of PEN were determined by BMD for all strains except for 17 *S. intermedius* isolates for which the MIC to PEN was obtained by gradient Etest because of insufficient growth in MH-F broth. The black vertical dotted line indicates the breakpoint for susceptibility in the context of infective endocarditis (0.125 mg/L), and the orange and red dotted lines indicate the EUCAST v 13.0 clinical breakpoints (0.25 and 2 mg/L). CLSI, Clinical and Laboratory Standards Institute; EUCAST, European Committee on Antimicrobial Susceptibility Testing.

intra-abdominal (41/93 episodes, 44.1%) and pulmonary (26/49 episodes, 53.1%) infections, whereas isolates from the *S. bovis/equinus*, *S. mitis*, and *S. sanguinis* groups were overrepresented in endocarditis (12/56, 21.4%; 18/56, 32.1%; and 19/56, 33.9%, respectively).

**Susceptibility to beta-lactams.** The MICs of benzylpenicillin (PEN), amoxicillin (AMX), cefotaxime (CTX), and ceftriaxone (CRO) were determined by BMD and gradient Etest for all isolates except 17 *S. intermedius* isolates for which MICs could only be measured by Etest because of insufficient growth in Muller-Hinton for Fastidious organisms (MH-F) broth. The agreement between BMD and Etest is shown in Fig. S1. The overall categorical agreement (average for all tested agents) was ≥ 94.9% (Table S2). A significant bias between the two techniques was observed for CTX and CRO for which MICs were lower with Etest in 24% and 67% of the isolates, respectively.

For the sake of clarity, only the BMD results of beta-lactam susceptibility are presented below, to which the Etest results of the 17 *S. intermedius* isolates with insufficient growth in MH-F broth are added. All isolates from the *S. anginosus*, *S. bovis/equinus*, and *S. mutans* groups were susceptible to PEN (Fig. 2) and to all the beta-lactam agents tested (Table 2). Conversely, 31.1% (46/148), 28.2% (11/39), and 51.9% (42/81) of the isolates from the *S. mitis*, *S. salivarius*, and *S. sanguinis* groups showed intermediate susceptibility or resistance to at least one beta-lactam, respectively (Fig. 3; Table 2). The different resistant phenotypes are shown in Table S3 and varied from resistance to a single beta-lactam agent to cross-resistance to all the beta-lactam agents tested. Overall, CTX remained the most frequently active agent, with resistance rates (MIC > 0.5 mg/L) of 18.2%, 2.5%, and 13.6% in the *S. mitis*, *S. salivarius*, and *S. sanguinis* group, respectively.

**Screening for beta-lactam resistance using disk diffusion.** The screening for beta-lactam resistance is based on DD testing of a 1-unit PEN disk where isolates showing an inhibition zone diameter ≥ 21 mm (European Committee on Antimicrobial Susceptibility Testing [EUCAST] Breakpoint Tables v. 13.0) can be categorized as susceptible to beta-lactam agents. We analyzed the performances of the screening for beta-lactam resistance with BMD as the reference method using this breakpoint and the one corresponding to the study-period (≥18 mm, EUCAST Breakpoint Tables v. 12.0). Using the 18 mm breakpoint (Fig. 4), 40/505 isolates (7.9%) were falsely categorized as susceptible to beta-lactams. The sensitivity, specificity, positive, and negative predictive values for the detection of beta-lactam resistance were 59.6%, 100%, 100%, and

**TABLE 2** Beta-lactam agents: susceptibility and MIC distributions of invasive non-beta-hemolytic *Streptococcus* isolates

| *Streptococcus* group (*n*; %) | Antimicrobial agent | MIC[a] (mg/L), *n* | | | | | | | | | | MIC interpretation[b], *n* (%) | | |
|---|---|---|---|---|---|---|---|---|---|---|---|---|---|---|
| | | ≤0.008 | 0.016 | 0.032 | 0.064 | 0.125 | 0.25 | 0.5 | 1 | 2 | ≥4 | S | I | R |
| *anginosus* | Benzylpenicillin | 13 | 23 | 18 | 92 | 26 | 0 | 0 | 0 | 0 | 0 | 172 (100%) | 0 | 0 |
| (172; 33.0%) | Ampicillin[c] | 4 | 3 | 17 | 31 | 51 | 47 | 2 | 0 | 0 | 0 | 155 (100%) | 0 | 0 |
| | Amoxicillin | 4 | 6 | 19 | 40 | 56 | 45 | 2 | 0 | 0 | 0 | 172 (100%) | 0 | 0 |
| | Cefotaxime | 0 | 4 | 10 | 13 | 41 | 95 | 9 | 0 | 0 | 0 | 172 (100%) | NA[d] | 0 |
| | Ceftriaxone | 0 | 2 | 6 | 11 | 27 | 54 | 72 | 0 | 0 | 0 | 172 (100%) | NA | 0 |
| *bovis/equinus* | Benzylpenicillin | 0 | 0 | 2 | 55 | 19 | 1 | 0 | 0 | 0 | 0 | 77 (100%) | 0 | 0 |
| (77; 14.8%) | Ampicillin | 0 | 0 | 0 | 14 | 62 | 1 | 0 | 0 | 0 | 0 | 77 (100%) | 0 | 0 |
| | Amoxicillin | 0 | 0 | 0 | 17 | 59 | 1 | 0 | 0 | 0 | 0 | 77 (100%) | 0 | 0 |
| | Cefotaxime | 0 | 0 | 1 | 20 | 52 | 4 | 0 | 0 | 0 | 0 | 77 (100%) | NA | 0 |
| | Ceftriaxone | 0 | 0 | 1 | 18 | 10 | 46 | 2 | 0 | 0 | 0 | 77 (100%) | NA | 0 |
| *mitis* | Benzylpenicillin | 12 | 17 | 28 | 14 | 15 | 20 | 11 | 9 | 5 | 17 | 106 (71.6%) | 25 (16.9%) | 17 (11.5%) |
| (148; 28.4%) | Ampicillin | 0 | 8 | 15 | 46 | 8 | 20 | 11 | 11 | 8 | 21 | 108 (73.0%) | 19 (12.8%) | 21 (14.2%) |
| | Amoxicillin | 1 | 10 | 25 | 37 | 17 | 16 | 4 | 11 | 8 | 19 | 110 (74.3%) | 19 (12.8%) | 19 (12.8%) |
| | Cefotaxime | 2 | 10 | 11 | 30 | 29 | 21 | 18 | 3 | 9 | 15 | 121 (81.8%) | NA | 27 (18.2%) |
| | Ceftriaxone | 0 | 5 | 13 | 14 | 41 | 17 | 23 | 10 | 7 | 18 | 113 (76.4%) | NA | 35 (23.6%) |
| *mutans* | Benzylpenicillin | 0 | 2 | 3 | 0 | 0 | 0 | 0 | 0 | 0 | 0 | 5 (100%) | 0 | 0 |
| (5; 1.0%) | Ampicillin | 0 | 0 | 0 | 5 | 0 | 0 | 0 | 0 | 0 | 0 | 5 (100%) | 0 | 0 |
| | Amoxicillin | 0 | 0 | 0 | 5 | 0 | 0 | 0 | 0 | 0 | 0 | 5 (100%) | 0 | 0 |
| | Cefotaxime | 0 | 0 | 2 | 3 | 0 | 0 | 0 | 0 | 0 | 0 | 5 (100%) | NA | 0 |
| | Ceftriaxone | 0 | 0 | 0 | 2 | 3 | 0 | 0 | 0 | 0 | 0 | 5 (100%) | NA | 0 |
| *salivarius* | Benzylpenicillin | 0 | 0 | 1 | 8 | 10 | 10 | 2 | 6 | 1 | 1 | 29 (74.4%) | 9 (23.1%) | 1 (2.6%) |
| (39; 7.5%) | Ampicillin | 0 | 1 | 0 | 5 | 10 | 9 | 6 | 5 | 2 | 1 | 31 (79.5%) | 7 (18.0%) | 1 (2.6%) |
| | Amoxicillin | 0 | 1 | 2 | 6 | 8 | 9 | 8 | 2 | 2 | 1 | 34 (87.2%) | 4 (10.3%) | 1 (2.6%) |
| | Cefotaxime | 1 | 0 | 6 | 8 | 9 | 8 | 6 | 1 | 0 | 0 | 38 (97.4%) | NA | 1 (2.6%) |
| | Ceftriaxone | 0 | 0 | 1 | 9 | 9 | 6 | 9 | 4 | 1 | 0 | 34 (87.2%) | NA | 5 (12.8%) |
| *sanguinis* | Benzylpenicillin | 4 | 8 | 0 | 1 | 21 | 15 | 14 | 10 | 6 | 2 | 49 (60.5%) | 30 (37.0%) | 2 (2.5%) |
| (81; 15.5%) | Ampicillin | 0 | 0 | 3 | 9 | 1 | 7 | 21 | 16 | 13 | 11 | 41 (50.6%) | 29 (35.8%) | 11 (13.6%) |
| | Amoxicillin | 0 | 0 | 0 | 10 | 3 | 9 | 20 | 20 | 14 | 5 | 42 (51.9%) | 34 (42.0%) | 5 (6.2%) |
| | Cefotaxime | 0 | 1 | 11 | 20 | 13 | 16 | 9 | 9 | 1 | 1 | 70 (86.4%) | NA | 11 (13.6%) |
| | Ceftriaxone | 0 | 0 | 6 | 10 | 26 | 10 | 13 | 8 | 7 | 1 | 65 (80.2%) | NA | 16 (19.8%) |

[a]MICs determined by broth microdilution except in the case of 17 *S. intermedius* isolates (*anginosus* group) for which MICs could only be measured by gradient Etest because of insufficient growth in MH-F broth.

[b]MICs interpretation according to EUCAST v 13.0. Susceptible isolates in shaded area, resistant isolates underlined.

[c]Ampicilllin MICs were not determined for 17 *S. intermedius* isolates (*anginosus* group), neither by broth microdilution because of insufficient growth in MH-F broth nor by gradient Etest.

[d]NA, not applicable; S, susceptible; I, susceptible increased exposure; R, resistant.

91.0%, respectively, with the lowest performances for the *S. sanguinis* group (Table S4). The failure to detect beta-lactam resistance was observed for a variety of phenotypes, including cross-resistance to all tested molecules (Table S5). Changing the breakpoint to 21 mm improved the detection of beta-lactam resistance, lowering false susceptibility to 4.2% (21/505) isolates and increasing sensitivity to 78.8% (Table S4 to S6).

**Susceptibility to macrolides, lincosamides, and streptogramins.** Susceptibility to MLS was studied using complementary methods based on DD and BMD. Susceptibility to clindamycin was measured by DD and BMD and inducible resistance to MLS was detected by DD according to EUCAST guidelines. Moreover, in the absence of specific European guidelines, susceptibility to macrolides was studied according to the guidelines of the French Society for Microbiology (24). The results of both methods showed some discrepancies regarding clindamycin susceptibility (Table 3; Fig. S2), as 28.5% of isolates (149/522) were found resistant using DD compared to 18.6% using BMD (94/505). Given that all resistant phenotypes observed by DD were confirmed at the genotypic level, DD testing was considered as the reference method for MLS susceptibility testing. Hence, the detection of MLS resistance using clindamycin MIC alone by BMD showed poor sensitivity, leading to very major errors in 9.9% of cases (50/505).

The resistance rates to MLS ranged from 30.8% in the *S. anginosus* group (53/172) to 68.8% in the *S. bovis/equinus* group (53/78). Resistant phenotypes and genotypes were differentially distributed among NBHS groups, the constitutive resistance to macrolides,

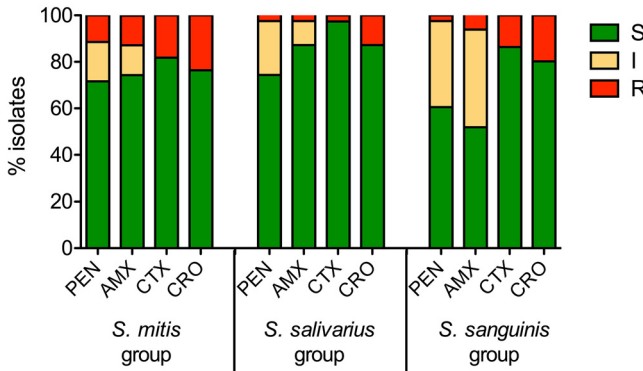

**FIG 3** Susceptibility to beta-lactam agents in streptococcal isolates of the *mitis*, *salivarius*, and *sanguinis* groups. Susceptibility to benzylpenicillin (PEN), amoxicillin (AMX), cefotaxime (CTX), and ceftriaxone (CRO) was measured by BMD. Results were interpreted according to EUCAST v13.0 clinical breakpoints. The number of isolates in each group was: *S. mitis*, n = 148; *S. salivarius*, n = 39; *S. sanguinis*, n = 81. EUCAST, European Committee on Antimicrobial Susceptibility Testing; S, susceptible; I, susceptible increased exposure; R, resistant.

lincosamides, and streptogramins B (cMLS$_B$ phenotype) being dominant in the *S. anginosus* and *S. bovis/equinus* groups (79% and 57%, respectively), whereas the sole resistance to macrolides (M phenotype) was the most frequent in the *S. mitis*, *S. salivarius*, and *S. sanguinis* groups (62%, 71%, and 57%, respectively; Table 3). Resistance to lincosamides and combined resistance to lincosamides and streptogramins A (L/LS$_A$ phenotype) was frequent among MLS resistant isolates of the *S. anginosus* and *S. bovis/equinus* groups (17% and 42%, respectively).

**Susceptibility to other antibiotics.** Table 4 and Fig. S2 summarize antimicrobial susceptibility testing by BMD to the other antibiotics. All the isolates displayed intrinsic low-level resistance to gentamicin with MICs ≤ 32 mg/L and were susceptible to vancomycin and teicoplanin (Table 4 and data not shown). Dalbavancin MIC distribution was modal and ranged from ≤ 0.004 to 0.125 mg/L, in contrast to that of rifampicin which was bimodal and ranged from ≤0.008 to ≥8 mg/L. Only 48.1% of the *S. bovis/equinus* isolates were reported as wild-type considering the EUCAST v 12.0 ECOFF at 0.125 mg/L. However, a one dilution-higher ECOFF at 0.25 mg/L (EUCAST v 13.0) categorizes all isolates but 11 (2.1%) as wild-type. Susceptibility to moxifloxacin was assessed by BMD and all but 8 isolates (1.6%) were reported as wild-type (ECOFF at 0.5 mg/L).

The frequency of tetracycline resistance, interpreted by DD according to the French guidelines in the absence of specific EUCAST guidelines (24), varied according to the streptococcal group, ranging from 7.7% in the *S. salivarius* group (3/39) to 70.1% in the *S. bovis/equinus* group (54/77) (Table 5). In the *S. mitis*, *S. salivarius*, and *S. sanguinis* groups, resistance was conferred almost exclusively by *tet*(M) whereas other determinant such as *tet*(L) and *tet*(O) were more common in the *S. anginosus* and *S. bovis/equinus* groups (Table 5).

Overall, multidrug-resistant isolates, defined as non-wild-type toward three or more classes of antibiotics, amounted to 4.8% (25/505), and included isolates that combined beta-lactam, clindamycin and tetracycline resistance.

**Analysis of NBHS isolates collected in 2022.** The specific study conducted in 2021 allowed us to collect and analyze more than 500 NBHS isolates. We sought to update these results through the analysis of the isolates sent on a regular basis to the Strep-NRC in 2022. A total of 79 NBHS invasive isolates were collected, which were addressed mainly for species identification and confirmation of specific resistance phenotypes. Despite the small size of this collection and the recruitment biases, we compared their characteristics to those of the 2021 collection.

The 79 isolates included 26 isolates of the *S. anginosus* group (33%), 21 isolates of the *S. bovis/equinus* (27%) and the *S. mitis* group (27%), seven isolates of the *S. sanguinis* group (8%), and four isolates of the *S. salivarius* group (5%). Their distribution and clinical characteristics seem similar to the 2021 collection (Table S7). All isolates of the *S. anginosus* and *S. bovis/equinus* groups remained susceptible to beta-lactam antibiotics as attested by

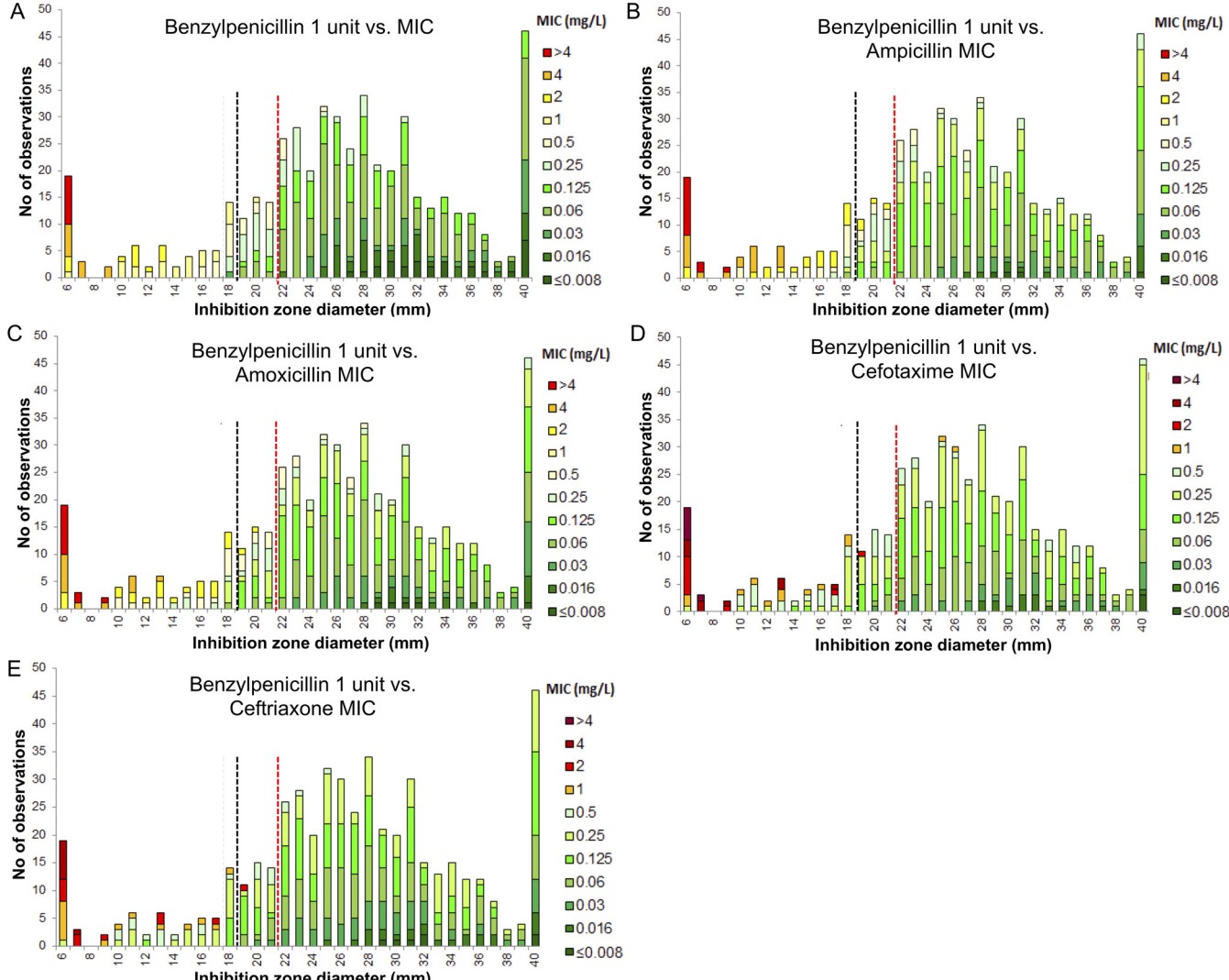

**FIG 4** Screening for beta-lactam resistance using the PEN 1U disk – predictive value for resistance to PEN (A), AMP (B), AMX (C), CTX (D), and CRO (E) as measured by MIC values determined by BMD in 505 NBHS invasive isolates. Screening cut-off values as recommended by EUCAST and a breakpoint of 18 mm (EUCAST guidelines v 12.0, black dotted line) and 21 mm (EUCAST guidelines v 13.0, red dotted line). Critical concentrations: PEN, 0.25 to 2 mg/L; AMP/AMX, 0.5 to 2 mg/L; CTX/CRO, 0.5 mg/L.

gradient tests (Table S8). Isolates of the *S. mitis* group displayed a higher rate of resistance than in 2021 (14/21, 67%) but most of the isolates were sent for confirmation of their resistance phenotype, some of them displaying particularly high MICs (>32 mg/L) for all tested molecules. Last, similar rates and patterns of resistance to MLS (Table S9) and tetracycline (Table S10) were observed between 2021 and 2022 for the different streptococcal groups. Resistance rates of the *S. salivarius* and *S. sanguinis* groups were not interpretable because of the small size of the collection.

## DISCUSSION

In this prospective survey, we studied more than 600 NBHS invasive isolates in France, including (i) an unbiased collection of 522 isolates responsible for invasive infections in 2021 from an *ad hoc* study conducted by the Strep-NRC and (ii) a collection of 79 isolates representing the total NBHS invasive isolates sent to the Strep-NRC by its correspondents in 2022 on a voluntary and routine basis, mainly for confirmation of species identification and antibiotic resistance phenotype. We gained a precise picture of NBHS infections and antimicrobial susceptibility at the species level for which the main findings are summarized below. The vast majority of cases occurred in the elderly, most often in male patients. Bacteremia without

**TABLE 3** Macrolides, lincosamides and streptogramins (MLS): susceptibility, resistance phenotypes and genotypes of invasive non-beta-hemolytic *Streptococcus* isolates

| Streptococcus group | Resistant isolates, n (%) | Disk diffusion phenotype[a], n (%) | | | | Clinda[Rb] n (%) | | Resistance determinants, n (%)[c] | | | | | |
|---|---|---|---|---|---|---|---|---|---|---|---|---|---|
| | | L/LS_A[g] | M | iMLS_B | cMLS_B | DD[d] | BMD[e] | mef | erm(A) | erm(B) | lnu(C) | lsa(A) | Others[f] |
| anginosus (n = 172) | 53 (30.8%) | 9 (17.0%) | 2 (3.8%) | 0 (0%) | 42 (79.2%) | 51 (29.7%) | 33 (21.3%) | 3 (5.7%) | 6 (11.3%) | 35 (66.0%) | 0 (0%) | 9 (17.0%) | 6 (11.3%) |
| bovis/equinus (n = 77) | 53 (68.8%) | 22 (41.5%) | 0 (0%) | 1 (1.9%) | 30 (56.6%) | 53 (68.8%) | 20 (26.0%) | 0 (0%) | 3 (5.7%) | 23 (43.4%) | 11 (13.4%) | 2 (3.8%) | 5 (9.4%) |
| mitis (n = 148) | 53 (35.8%) | 2 (3.8%) | 33 (62.3%) | 2 (3.8%) | 16 (30.2%) | 20 (37.8%) | 18 (34.0%) | 38 (71.7%) | 6 (11.3%) | 11 (20.8%) | 0 (0%) | 0 (0%) | 1 (1.9%) |
| mutans (n = 5) | 0 | NA | NA | NA | NA | 0 | 0 | NA | NA | NA | NA | NA | NA |
| salivarius (n = 39) | 17 (43.6%) | 0 (0%) | 12 (70.6%) | 0 (0%) | 5 (29.4%) | 5 (12.8%) | 4 (10.3%) | 13 (76.5%) | 0 (0%) | 5 (29.4%) | 0 (0%) | 0 (0%) | 0 (0%) |
| sanguinis (n = 81) | 46 (56.8%) | 1 (2.2%) | 26 (56.5%) | 0 (0%) | 19 (41.3%) | 20 (24.7%) | 19 (23.5%) | 28 (60.9%) | 4 (8.7%) | 15 (32.6%) | 0 (0%) | 1 (2.2%) | 1 (2.2%) |
| Total[e] (n = 522) | 222 (42.5%) | 34 (15.3%) | 73 (32.9%) | 3 (1.4%) | 112 (50.7%) | 149 (28.5%) | 94 (18.6%) | 82 (36.9%) | 19 (8.6%) | 89 (40.1%) | 11 (5.0%) | 12 (5.4%) | 13 (5.9%) |

[a]DD phenotypes were interpreted according to the guidelines on Antibiotic Susceptibility Testing of the French Society for Microbiology (CA-SFM).

[b]Minimum inhibitory concentrations of clindamycin were measured by BMD and interpreted according to EUCAST clinical breakpoint table v 13.0.

[c]Percentages are relative to total resistant isolates within each group.

[d]Including the L/LS_A, iMLS_B and cMLS_B phenotypes.

[e]A total of 17 *S. intermedius* isolates could not be tested by BMD but only by DD because of insufficient growth in MH-F broth. These included 5 MLS resistant isolates of which 4 cMLS_B phenotypes and 1 L phenotype.

[f]Including *erm* (T) (5 in the *bovis/equinus* group), *lnu* (A) (2 in the *anginosus* group), *lnu* (D) (1 in the *mitis* group), *lnu* (E) (1 in the *anginosus* group), and *lsa* (B) (3 in the *anginosus* group and 1 in the *sanguinis* group).

[g]Including *erm* (T) (5 in the *bovis/equinus* group), *lnu* (A) (2 in the *anginosus* group); Clinda[R], clindamycin resistance; DD, disk diffusion; L, resistance to lincosamides; LS_A, resistance to lincosamides and streptogramin A; M, efflux-mediated resistance to C14 and C15 macrolides; iMLSB, inducible resistance to macrolides, lincosamides, and streptogramin B; cMLS_B, constitutive resistance to macrolides, lincosamides, and streptogramin B; NA, not applicable.

**TABLE 4** Non-beta-lactam agents: susceptibility and MIC range of invasive non-beta-hemolytic *Streptococcus* isolates

| *Streptococcus* group (n, %) | Antimicrobial agent | MIC (mg/L) Range | MIC interpretation (%) | |
| --- | --- | --- | --- | --- |
| | | | S/wild-type[a,b] | R/non-wild-type[a] |
| *anginosus* (155, 30.7%) | Dalbavancin | ≤0.004 to 0.064 | 100 | 0 |
| | Moxifloxacin | ≤0.032 to 0.25 | 100 | 0 |
| | Rifampicin | ≤0.008 to ≥8 | 96.1 | 3.9 |
| | Vancomycin | 0.25 to 1 | 100 | 0 |
| *bovis/equinus* (77, 15.2%) | Dalbavancin | 0.008 to 0.125 | NA | NA |
| | Moxifloxacin | 0.125 to ≥8 | 97.4 | 2.6 |
| | Rifampicin | 0.032 to 0.25 | 100 | 0 |
| | Vancomycin | 0.25 to 0.5 | 100 | 0 |
| *mitis* (148, 29.3%) | Dalbavancin | ≤0.004 to 0.125 | NA | NA |
| | Moxifloxacin | 0.064 to 4 | 98.6 | 1.4 |
| | Rifampicin | 0.016 to ≥8 | 98.6 | 1.4 |
| | Vancomycin | 0.25 to 1 | 100 | 0 |
| *mutans* (5, 1.0%) | Dalbavancin | 0.032 to 0.06 | NA | NA |
| | Moxifloxacin | 0.125 | 100 | 0 |
| | Rifampicin | 0.016 to 0.032 | 100 | 0 |
| | Vancomycin | 0.5 to 1 | 100 | 0 |
| *salivarius* (39, 7.7%) | Dalbavancin | ≤0.004 to 0.125 | NA | NA |
| | Moxifloxacin | ≤0.032 to 0.5 | 100 | 0 |
| | Rifampicin | 0.032 to ≥8 | 89.7 | 10.3 |
| | Vancomycin | 0.25 to 1 | 100 | 0 |
| *sanguinis* (81, 16.0%) | Dalbavancin | ≤0.004 to 0.06 | NA | NA |
| | Moxifloxacin | 0.066 to ≥8 | 95.1 | 4.9 |
| | Rifampicin | 0.032 to ≥8 | 96.3 | 3.7 |
| | Vancomycin | 0.25 to 1 | 100 | 0 |

[a]Dalbavancin, and vancomycin were interpreted according to EUCAST clinical breakpoints v 13.0 (S or R). Moxifloxacin and rifampicin were interpreted as wild-type or non-wild-type in accordance with EUCAST ECOFFs (https://mic.eucast.org/; reviewed September 2022).
[b]S, susceptible; R, resistant; NA, not applicable (no breakpoint).

focus, intra-abdominal infections, and endocarditis accounted for 75% of the cases. Although all *viridans* streptococcal groups were involved, 85% of the cases were caused by not more than 10 species, with significant associations between specific clinical manifestations and NBHS groups. Susceptibility to antimicrobial agents was highly variable among NBHS isolates. Because the collection mode over the two study-periods was different, and because the isolates analyzed in the second study-period are very unlikely to be representative of the population of human infective isolates, analyzing the evolutionary trends regarding antibiotic susceptibility would not be reliable. Therefore, only the general features highlighted by the 2021 collection are discussed. Overall, whereas all isolates of the *S. anginosus* and *S. bovis/equinus* groups were susceptible to beta-lactams, intermediate susceptibility and resistance were encountered in approximately 30% isolates of the *S. mitis* and *S. salivarius* groups and in more than 50% isolates of the *S. sanguinis* group. Resistance to MLS was also very common among all streptococcal groups, always exceeding 30% of isolates.

**TABLE 5** Tetracyclines: susceptibility and resistance determinants of invasive non-beta-hemolytic *Streptococcus* isolates[a]

| *Streptococcus* group | Resistant isolates, n (%) | Resistance determinant, n (%)[b] | | | | |
| --- | --- | --- | --- | --- | --- | --- |
| | | *tet*(M) | *tet*(L) | *tet*(O) | *tet*(T) | Unknown |
| *anginosus*, n = 172 | 52 (30.2%) | 42 (80.8%) | 0 (0%) | 7 (13.5%) | 1 (1.9%) | 2 (3.8%) |
| *bovis/equinus*, n = 77 | 54 (70.1%) | 37 (68.5%) | 18 (33.3%) | 16 (20.8%) | 0 (0%) | 0 (0%) |
| *mitis*, n = 148 | 24 (16.2%) | 23 (95.8%) | 0 (0%) | 1 (4.2%) | 0 (0%) | 0 (0%) |
| *mutans*, n = 5 | 0 (0%) | 0 (0%) | 0 (0%) | 0 (0%) | 0 (0%) | 0 (0%) |
| *salivarius*, n = 39 | 3 (7.7%) | 3 (100%) | 0 (0%) | 0 (0%) | 0 (0%) | 0 (0%) |
| *sanguinis*, n = 81 | 25 (30.9%) | 24 (96.0%) | 0 (0%) | 0 (0%) | 0 (0%) | 1 (0%) |
| Total, n = 522 | 158 (30.3%) | 129 (81.6%) | 18 (11.4%) | 24 (15.2%) | 1 (0.6%) | 3 (1.9%) |

[a]Interpreted according to the guidelines on Antibiotic Susceptibility Testing of the French Society for Microbiology.
[b]Percentages are relative to total resistant isolates within each group.

Comprehensive studies on NBHS invasive infections are scarce. Still, regarding the clinical characteristics, our study confirms the previously described associations between the *S. anginosus* group with abscesses and pleural fluid infections (10, 12), together with the very high risk-association (>30%) between *S. bovis/equinus* and *S. sanguinis* groups bloodstream infections with endocarditis, illustrating the representativeness of our collection (25, 26). Our results also underline an increased susceptibility of male patients and of the elderly to NBHS infections, in agreement with specific reports on invasive *S. anginosus* group infections whose incidence increases with age and whose gender ratio usually exceeds 1.5 (11–13). These observations can be linked to the higher prevalence of NBHS infections in adults suffering from underlying diseases which are overrepresented in the elderly (6) and to the overrepresentation of male patients in endocarditis which might account for the overrepresentation of males in NBHS infective endocarditis. Nevertheless, the increased susceptibility of male patients was observed in all clinical manifestations of NBHS infections and whatever streptococcal group considered, reflecting similar gender and age susceptibility as that described for *S. pneumoniae* infections (27).

Considering susceptibility to antibiotics, resistance rates to beta-lactams and MLS among NBHS species were similar to those reported elsewhere (19, 28, 29). Resistance rates to beta-lactams of 30% were described in the United States among *S. mitis* isolates in a study including over 4,800 *S. mitis*, *S. oralis*, and nonspeciated NBHS isolates from bloodstream infections between 2010 and 2020 (28). Resistance rates were stable over the 10-year study period. Marron et al. also found a ceftriaxone resistance rate over 30% in NBHS strains, including 80% *S. mitis* strains in isolates from blood cultures of neutropenic patients with cancer (30). In agreement with our results, higher resistance rates among isolates of the *S. salivarius* and *S. sanguinis* groups were also previously described, reaching up to 70% isolates (30). In contrast to these high rates of beta-lactam resistance, all strains remained susceptible to vancomycin as described by others and dalbavancin activity was in agreement with previous studies on nonspeciated NBHS isolates (28, 30–32). In our study, none of the isolates displayed high-level resistance to gentamicin, also in agreement with other studies where high-level resistance rates were below 2% (30). Concerning MLS and tetracycline, previous studies on NBHS isolates from diverse flora and streptococcal groups reported resistance as common (>10%), with the highest resistance rates among the *S. bovis/equinus* group (12, 13, 28, 29, 33, 34).

Our study emphasizes the critical impact of recommendations for the interpretation of antibiotic susceptibility testing and the efforts needed for their worldwide harmonization. France and most European countries, including Germany, the Netherlands, Norway, Sweden, United Kingdom, and others agreed to create common European guidelines known as the EUCAST guidelines. These are followed by many countries outside Europe such as Australia, Brazil, China, Morocco, and South Africa. On the other hand, the United States, Canada, and other countries follow the Clinical and Laboratory Standard Institute (CLSI) guidelines (35). CLSI and EUCAST cooperate more and more frequently through dedicated study groups but have no common clinical breakpoint process and antibiotic susceptibility testing interpretation might be different when using one guideline or another. For instance, interpretation criteria for some antimicrobials might be missing in the EUCAST guidelines and present in the CLSI guidelines. In such cases, the French guidelines might refer to previous national guidelines or to the CLSI guidelines (36), which was the case in our study for the interpretation of NBHS susceptibility to erythromycin and tetracycline. Moreover, the clinical breakpoints and the screening methods for the detection of resistance might also differ, as exemplified by beta-lactam susceptibility testing for NBHS. In the EUCAST guidelines, screening for decreased susceptibility and resistance to beta-lactams relies on the PEN 1 unit disk, and we demonstrated that it was improved with the recent change from ≥18 to ≥21 mm in the EUCAST Breakpoint Tables. Conversely, the CLSI guidelines specify that NBHS isolated from normally sterile sites should be tested for PEN susceptibility using an MIC method. In our study, the screening for beta-lactam resistance using PEN 1 unit disk remained poorly sensitive despite the change introduced in the v 13.0 Breakpoint Table, also

emphasizing the need for accurate testing by MIC measurements in invasive and difficult-to-treat infections due to NBHS.

Overall, this study provides a detailed analysis at the species level of the epidemiology of NBHS infections in France. Notable strengths are (i) the representativeness of our cohort, (ii) the exhaustive analysis of the isolates at the species level and the inclusion of a wide variety of antimicrobial agents for antibiotic susceptibility testing and (iii) the combination of multiple techniques for antibiotic susceptibility testing. First of all, the French Strep-NRC network includes more than 200 laboratories located throughout France, which brings a nationwide coverage to the study. Correspondents were asked to send all NBHS isolates isolated from normally sterile sites without any selection during a 2-month study period in 2021, in order to ensure maximum completeness of the collection. Second, all isolates were identified at the species level, assigned to a streptococcal group, and antibiotic susceptibility testing included 5 beta-lactam molecules, MLS, three glycopeptides, and four more antimicrobial agents. Last, antibiotic susceptibility testing was performed using complementary techniques, including BMD and gradient test for beta-lactams, and BMD and DD for MLS, thus reinforcing the robustness of the results. Besides, microbiological analyses were performed in a unique laboratory, discrepant results were controlled in two independent laboratories and resistance to MLS and tetracycline was confirmed by the identification of resistance genes. However, some limitations must be considered. The initial study was carried out over a short period of 2 months, which possibly introduced biases regarding the clinical epidemiology and the species collected, all the more so as it coincided with containment measures related to the COVID-19 epidemic in France. Notably, the number of isolates among the *mutans* group was very limited. We added to this study the isolates sent in 2022 by our correspondents on a routine basis, mainly for expertise regarding species identification and antibiotic resistance profiles, but the number of invasive isolates collected was very limited, disabling any comparison between the two periods. Considering antimicrobial susceptibility, the genetic supports of resistance to beta-lactams and moxifloxacin were not investigated, but will be the subject of future studies relying on whole-genome sequencing.

Altogether, this study highlights the wide spectrum of diseases caused by NBHS and the close relationship between specific groups, clinical manifestations, and antimicrobial resistance profiles. Comparing the resistance profiles and the genetic determinants between NBHS groups allowed us to identify both distinct and shared patterns that reflect common ecological niches and relationships with human and animal reservoirs, such as the high prevalence of resistance to beta-lactams and efflux-mediated resistance to macrolides in streptococcal species of the oral cavity (*S. mitis*, *S. salivarius*, and *S. sanguinis* groups) (29, 33) and of cMLS$_B$ and L/LS$_A$ phenotypes in streptococcal species of the intestinal tract (*S. anginosus* and *S. bovis*/*equinus* groups) (34). Importantly, our findings shed the light on the extremely high prevalence of resistance to beta-lactams in NBHS species of the oral cavity, which can furthermore display unusual phenotypes which might not be detected by benzylpenicillin testing alone. Therefore, accurate antibiotic susceptibility testing of beta-lactams by MICs measurements seems mandatory for the treatment of NBHS invasive infections. Besides, the possible emergency of beta-lactam resistance in NBHS of the intestinal tract and of multidrug-resistant isolates in NBHS requires a continuous surveillance of NBHS susceptibility to antimicrobials.

In conclusion, despite limitations mainly regarding the short 2-month study period, this work provides a first accurate overview of the epidemiology of NBHS invasive infections in France and will serve as a basis for future studies aiming at monitoring NBHS epidemiology. This work highlights the high level of beta-lactam nonsusceptibility in invasive streptococcal isolates belonging to the *S. mitis*, *S. salivarius*, and *S. sanguinis* groups, provides evidence of the diversity of antibiotic susceptibility patterns among NBHS, and emphasizes the need for a continuous surveillance of NBHS epidemiology and antibiotic susceptibility.

## MATERIALS AND METHODS

**Isolates and data collection.** Invasive isolates of NBHS were prospectively referred to the French Strep-NRC (https://cnr-strep.fr) by correspondents located throughout France. Participants were asked to send all

invasive NBHS (excluding pneumococci) isolated between March and April 2021. Besides, isolates are regularly sent on a routine basis by the correspondents for expertise, mainly for confirmation of species identification and study of antibiotic resistance. Clinical data (age, sex, clinical manifestations) were retrieved from questionnaires. Only isolates responsible for invasive infections, e.g., isolated from a normally sterile site, were considered.

Isolates were identified by MALDI-TOF mass spectrometry (Microflex; Bruker Daltonics, Bremen, Germany) using a local database enriched with NBHS spectra which includes 20, 30, and seven reference spectra for the difficult-to-identify species *S. mitis*, *S. oralis*, and *S. pseudopneumoniae*. Species confirmation was performed using sequencing of the *sodA* gene when mass spectrometry identification was ambiguous (23). Isolates which could not be identified at the species level were excluded from the analysis.

**Antibiotic susceptibility testing.** MIC of PEN, AMP, AMX, CTX, CRO, gentamicin, clindamycin, rifampicin, moxifloxacin, vancomycin, and dalbavancin were determined by BMD for NBHS isolated between March and April 2021 using a custom microtiter plate (SWE1SPEC, Thermo Fisher Scientific, Basingstoke, UK) and in-house prepared broth in the form of MH-F according to EUCAST instructions (37). MIC of PEN, AMX, CTX, and CRO were also determined by gradient tests (Etest, bioMérieux, Marcy-l'Étoile, France) on MH-F agar (bioMérieux, Marcy-l'Étoile, France) for all NBHS isolates.

Susceptibility to PEN, AMP, gentamicin, erythromycin, clindamycin, tetracycline, vancomycin, and teicoplanin was also determined by DD (bioMérieux, Marcy-l'Étoile, France) according to EUCAST guidelines (http://www.eucast.org) for all NBHS isolates.

Quality controls using *S. pneumoniae* ATCC 49619 were performed weekly. Determinants for resistance to MLS and tetracyclines were detected by multiplex PCR (38, 39).

**Data interpretation and analysis.** Antibiotic susceptibility testing results were interpreted according to EUCAST Breakpoint Tables v. 13.0 unless otherwise specified (40, 41). When no breakpoints were available, data were interpreted according to epidemiological cut-off values (ECOFFs) or to the Committee on Antimicrobial Susceptibility Testing of the French Society for Microbiology (CA-SFM) (24). For beta-lactam susceptibility testing, results of Etests and DD were compared to reference BMD. Discrepant results and unusual phenotypes were double-checked in two independent laboratories. To allow comparison with BMD, Etest values were rounded up to the nearest 2-fold dilution. Very major errors, major errors, and minor errors were calculated as described by the Clinical and Laboratory Standards Institute (42). Overall categorical agreement against BMD reference MICs was evaluated for DD and Etest interpretations. Absolute and essential agreement, as well as the bias, were calculated for Etest according to ISO 20776-2 (43).

**Statistical analysis.** Categorical variables are expressed as numbers and percentages. Statistical analyses were performed using chi-square test. A *P*-value <0.05 was regarded as significant.

## SUPPLEMENTAL MATERIAL

Supplemental material is available online only.
**SUPPLEMENTAL FILE 1**, PDF file, 3.7 MB.
**SUPPLEMENTAL FILE 2**, PDF file, 3.7 MB.
**SUPPLEMENTAL FILE 3**, PDF file, 0.2 MB.

## ACKNOWLEDGMENTS

We thank all the correspondents of the French National Reference Center for Streptococci.

Part of this work has been presented at the 41st RICAI annual meeting, Paris, France and at the 32nd ECCMID annual meeting, Lisbon, Portugal.

This work was partially supported by EUCAST, Assistance Publique – Hôpitaux de Paris and Santé publique France.

We declare no conflicts of interest.

Writing – original draft: A.T. and C.Pl.; writing – review and editing: A.T., C.Pl., G.K., and E.M.; conceptualization: A.T., C.Pl., and C.Po.; methodology: A.T., C.Pl., G.K., and E.M.; data acquisition: A.F., C.Pl., E.B., N.D., M.G., and Y.B.; data curation: A.F., C.Pl., N.D., and M.G.; supervision: A.T.; funding acquisition: A.T., C.Pl., and C.Po.

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
