## [Reviewer comments · Microbiology Spectrum]

Microbiology Spectrum

Microbiological epidemiology of invasive infections due to non-beta-hemolytic streptococci, France, 2021

Céline Plainvert, Erika Matuschek, Nicolas Dmytruk, Marine Gaillard, Amandine Frigo, Yassine Ballaa, Eddy Biesaga, Gunnar Kahlmeter, Claire Poyart, and Asmaa Tazi

Corresponding Author(s): Asmaa Tazi, Assistance Publique - Hopitaux de Paris

Review Timeline:

Submission Date:	January 13, 2023
Editorial Decision:	February 28, 2023
Revision Received:	April 15, 2023
Accepted:	April 29, 2023

Editor: Po-Yu Liu

Reviewer(s): The reviewers have opted to remain anonymous.

Transaction Report:

DOI: <https://doi.org/10.1128/spectrum.00160-23>

February 28, 2023

Dr. Asmaa Tazi
Assistance Publique - Hopitaux de Paris
Paris
France

Re: Spectrum00160-23 (Microbiological epidemiology of invasive infections due to non-beta-hemolytic streptococci, France, 2021)

Dear Dr. Asmaa Tazi:

Thank you for submitting your manuscript to Microbiology Spectrum. The article presents a significant number of results with 522 invasive NBHS and shows promise for making a valuable contribution to the field. However, the reviewer has noted concerns about the sampling period in 2021, which may not reflect current knowledge on the topic. To enhance the quality and relevance of your work, I encourage you to revise and update the article to reflect current knowledge and insights. When submitting the revised version of your paper, please provide (1) point-by-point responses to the issues raised by the reviewers as file type "Response to Reviewers," not in your cover letter, and (2) a PDF file that indicates the changes from the original submission (by highlighting or underlining the changes) as file type "Marked Up Manuscript - For Review Only". Please use this link to submit your revised manuscript - we strongly recommend that you submit your paper within the next 60 days or reach out to me. Detailed instructions on submitting your revised paper are below.

Link Not Available

Sincerely,

Po-Yu Liu

Journals Department
Reviewer comments:

Reviewer #1 (Comments for the Author):

I believe that this is a very informative study on a current snapshot of invasive non-hemolytic streptococcal isolates.

1. A somewhat major criticism is that I wish that the authors would have included the under-studied beta hemolytic non group A and non group B streptococcal species in this very well performed study.
2. A rather minor criticism: Although I believe that the bacteriologic methods were expertly performed, I cannot help but wonder about the species identification of *S. mitis* and *S. oralis*. At least in my somewhat limited experience, I have found that

identification/separation of *S. mitis*, *S. oralis*, and *S. pseudopneumoniae* can be quite difficult phenotypically and on the basis of a single targeted sequence. On the other hand, considerable experience with some sort of phylogenetic analysis involving either multiple gene targets (ie MLST loci) or whole genomes is be more straightforward to me. This all said, this is a very useful study in a very neglected area.

Reviewer #2 (Comments for the Author):

This work by Plainvert and colleagues is a straight-forward description and analysis of the species composition and antibiotic-resistance profiles of non-beta hemolytic streptococci isolated from invasive disease over a several month period in France. The presentation is very clear and logical, the analysis of resistance mechanisms is comprehensive and technically well-done. The large number of strains makes the study robust and the comparison of different methods for determination of resistance profile will be of interest. Comments below are minor and address areas that could be made more clearly for the reader.

- 1.
2. Line 102. Here or in the methods, include a reference for the soda sequencing method.
3. Lines 147-148. This is confusing. Comparisons are made and are concluded as "false" or "incorrect". But the comparison used to make this conclusion is not clearly stated. What is the "gold standard (reference standard)" assay. And it does not seem that a gold standard assay is consistently applied between the multiple comparisons that are made, sometimes it is the BMD assay and sometimes the DD assay. A clearly presentation of these comparisons would be useful.
4. Related to the above, should not the gold standard for these comparisons be the genotypic analysis?
5. Often results are interpreted using as criteria the "French Standards." Generalization of these conclusions outside of France would benefit from some discussion of similarities and differences between these standards and standards used in other regions.

Reviewer #3 (Comments for the Author):

Manuscript Number: Spectrum00160-23

The manuscript entitled "Microbiological epidemiology of invasive infections due to non-beta-hemolytic streptococci, France, 2021" by Céline Plainvert et al., describes the clinical and microbiological epidemiology of invasive infections due to non-beta-hemolytic streptococci during the period March and April 2021, along with the screening of their behavior towards different antimicrobial agents.

The paper reported a relevant issue, however many weakness are highlighted into the text.

Major points to address:

- 1) Even if the authors sampled 522 invasive NBHS between March and April 2021; I really believe that it is too dated since, until now, additional period should have been considered to strengthen the results. In the manuscript the authors identified and characterized bacteria of 2 years ago, considering the speed at which bacteria acquire resistance, the picture they report is not current at all. This is a crucial limitation of the study.
- 2) The introduction section is too generic. The authors did not did not deepen the numerical relevance of the impact of the following sentences: "These commensals are responsible for a wide range of invasive diseases, particularly in the presence of underlying conditions and in immune-compromised adults. Importantly, NBHS are reported as the first or second cause of infective endocarditis worldwide [3-5]. Additionally, isolates of the *Streptococcus anginosus* group are causative agents of pyogenic infections and abscesses in immunocompetent adults and children [6]."
- 3) In the discussion, the authors present only few comparisons with literature data; the same concepts are repeated trough the text without creating a logical and continuous argumentation. The last paragraph of the manuscript in addition to the overall conclusion of the research should contain the limitations of the study. The author should rewrite the discussion to increase its overall quality.

Minor points

Please check all the text making attention in abbreviations (for that reason it should be in extenso the first time mentioned and then the abbreviation) and in bacterial nomenclature (i.e. prefer the term *S. mitis* instead of mitis, and write *Streptococcus* spp. when appropriate).

Staff Comments:

Preparing Revision Guidelines

Please return the manuscript within 60 days; if you cannot complete the modification within this time period, please contact me. If you do not wish to modify the manuscript and prefer to submit it to another journal, please notify me of your decision immediately so that the manuscript may be formally withdrawn from consideration by Microbiology Spectrum.

Spectrum00160-23 - Microbiological epidemiology of invasive infections due to non-beta-hemolytic streptococci, France, 2021

Response to reviewers

First of all, we would like to thank the reviewers for their positive comments and constructive feedback which hopefully helped us improve the overall quality of the manuscript.

Below are the specific responses to their questions.

- **Reviewer #1**

I believe that this is a very informative study on a current snapshot of invasive non-hemolytic streptococcal isolates.

- We thank the reviewer for his positive comments.

Question 1. A somewhat major criticism is that I wish that the authors would have included the understudied beta hemolytic non-group A and non-group B streptococcal species in this very well performed study.

- We fully understand the reviewer's willingness to have details on beta hemolytic non-group A and non-group B streptococcal species, which mainly include group C and G streptococci of the human species *Streptococcus dysgalactiae* subsp. *equisimilis* and the animal species *Streptococcus canis* and *Streptococcus equi*. Nevertheless, these species differ from non-beta-hemolytic streptococci (NBHS) in many ways: they are considered as pathogens, are more often responsible for invasive diseases, and far less resistant to antibiotics, especially to beta-lactam agents. We are currently preparing an article dedicated to their clinical and microbiological epidemiology based on the data collected by the French National Reference Center for streptococci over the past 10 years (similar to what our team previously published, Loubinoux et al, J Clin Microbiol 2013, doi: 10.1128/JCM.01262-13). This article should be submitted by the summer 2023,.

In the article submitted to Microbiology Spectrum , we aimed to focus on NBHS, as specified in the title, to give a precise overview of their features without diluting the main

messages and we would prefer to keep this initial objective. We hope that these explanations will be satisfactory.

Question 2. A rather minor criticism: Although I believe that the bacteriologic methods were expertly performed, I cannot help but wonder about the species identification of *S. mitis* and *S. oralis*. At least in my somewhat limited experience, I have found that identification/separation of *S. mitis*, *S. oralis*, and *S. pseudopneumoniae* can be quite difficult phenotypically and on the basis of a single targeted sequence. On the other hand, considerable experience with some sort of phylogenetic analysis involving either multiple gene targets (ie MLST loci) or whole genomes is more straightforward to me. This all said, this is a very useful study in a very neglected area.

- The reviewer is absolutely right and the separation of the streptococcal species *S. mitis*, *S. oralis* and *S. pseudopneumoniae* is somehow difficult. Our identification is based on MALDI-tof (Bruker Daltonics) using an in-house database. Indeed, we enriched the Bruker database with our collection of streptococcal isolates which were identified using *sodA* and *16S rDNA* sequencing. The local database used for species identification includes 30, 20, and 7 spectra for *S. mitis*, *S. oralis*, and *S. pseudopneumoniae*, respectively. Identification to the species level is defined as accurate when the 10 proposals (7 first for *pseudopneumoniae*) are concordant. In the other cases, identification is confirmed by *sodA* and *16S rDNA* sequencing. Isolates for which species identification was inconsistent or undetermined were excluded from the dataset presented in the article. Therefore, we believe that the results presented here are reliable.

A sentence specifying the number of reference spectra for these 3 species was added to the Methods section lines 370-372.

- **Reviewer #2**

This work by Plainvert and colleagues is a straight-forward description and analysis of the species composition and antibiotic-resistance profiles of non-beta hemolytic streptococci isolated from invasive disease over a several month period in France. The presentation is very clear and logical, the analysis of resistance mechanisms is comprehensive and technically well-done. The large number of strains makes the study robust and the comparison of

different methods for determination of resistance profile will be of interest. Comments below are minor and address areas that could be made more clearly for the reader.

Question 1. Line 102. Here or in the methods, include a reference for the *sodA* sequencing method.

- A reference for the *sodA* sequencing method was already included in the method section: reference 15 in the previous version of the manuscript, reference 23 in the revised version: Poyart C, Quesne G, Coulon S, Berche P, Trieu-Cuot P. Identification of streptococci to species level by sequencing the gene encoding the manganese-dependent superoxide dismutase. J Clin Microbiol 1998;36:41–7.

For clarity, this reference has now been added at first mention of the *sodA* sequencing method, in the results section line 116.

Question 2. Lines 147-148. This is confusing. Comparisons are made and are concluded as "false" or "incorrect". But the comparison used to make this conclusion is not clearly stated. What is the "gold standard (reference standard)" assay. And it does not seem that a gold standard assay is consistently applied between the multiple comparisons that are made, sometimes it is the BMD assay and sometimes the DD assay. A clearly presentation of these comparisons would be useful.

- The reviewer is right and we apologize for this lack of clarity. The gold standard used for beta-lactam susceptibility testing was the BMD test whereas the gold standard for macrolides, lincosamides, and streptogramins (MLS) was disk diffusion (DD) for a better detection of the inducible resistance as specified in the EUCAST guidelines. This issue has been clarified in the results section line 161, where we specified “with BMD as the reference method” for beta-lactam susceptibility testing, and line 180 “DD testing was considered the reference method for MLS susceptibility testing”. These details were also added to the methods section lines 395-397 “For beta-lactam susceptibility testing, results of Etests and DD were compared to reference BMD.”

The choice of DD as the reference method for susceptibility to macrolides, lincosamides and streptogramins (MLS) was explained in the first version of the manuscript lines 156-160 by the following sentence: “Results of DD and BMD testing regarding clindamycin susceptibility showed some discrepancies (...) All resistant phenotypes observed by DD were

confirmed at the genotypic level. DD testing was therefore considered as the reference method...” and we tried to make it clearer in this revised version.

Question 3. Related to the above, should not the gold standard for these comparisons be the genotypic analysis?

- Using DD as the reference method for MLS could be a possibility given that resistance determinants are easily detectable. Unfortunately, the situation is more complex in the case of beta-lactam susceptibility, for which resistance is conferred by point mutations in *pbp* encoding genes. Although some typical point mutations conferring resistance to beta-lactam agents have been identified, especially in *Streptococcus pneumoniae*, these are not so well-defined in non-beta-hemolytic streptococci.

Besides, genotypic analysis considering resistance to both MLS and beta-lactams includes several drawbacks and limitations: i) inferring a phenotype from a genotype is not as reliable and robust as one would expect, ii) using a genotypic analysis on this collection of 522 isolates would imply to get the whole genome sequence of all the isolates which would be time and money consuming, and iii) most importantly, there is no recommendations for genotypic data interpretation in the context of antibiotic susceptibility testing. Therefore, we think that keeping phenotypic analyses and BMD as the gold standard for antibiotic susceptibility testing and beta-lactam susceptibility testing remains relevant.

Question 4. Often results are interpreted using as criteria the "French Standards." Generalization of these conclusions outside of France would benefit from some discussion of similarities and differences between these standards and standards used in other regions.

- We apologize for the lack of clarity about the interpretation criteria and a few explanations are provided below.

In Europe, there is actually only one guideline – all the national committees have signed "contracts" with EUCAST where they abandon the national guidelines (Sweden, Norway, France, Netherlands, Germany, and the UK) in exchange for equal influence in the creation of a European common guideline. For practical purposes and for posting national translations they keep their identities and websites but have agreed to follow EUCAST guidelines.

The USA follow the Clinical and Laboratory Standard Institute (CLSI) guidelines. CLSI and EUCAST cooperate more and more frequently, but we have no common breakpoint

process and neither has CLSI and FDA. So internationally, there are EUCAST guidelines, CLSI guidelines and FDA guidelines.

Outside Europe and the USA, many countries have signed up to the EUCAST guidelines (Brazil, New Zealand, South Africa, Australia, China, Morocco, Algeria, to mention some).

In France, we follow the EUCAST guidelines except for some specific molecules for which EUCAST guidelines are missing and for which previous national guidelines or CLSI guidelines are applied. Considering our article, the results interpreted according to the French criteria consist in those of erythromycin and tetracycline and the corresponding clinical breakpoints are practically identical to those of the CLSI.

We added these explanations in the results section, specifying that French guidelines were used in the absence of European guidelines, lines 176-177 for MLS and 205 for tetracycline. We also added a paragraph to the discussion section and argued a bit more about the complex issue of worldwide harmonization of antibiotic susceptibility testing interpretation criteria, lines 287-310.

- **Reviewer #3**

Manuscript Number: Spectrum00160-23

The manuscript entitled "Microbiological epidemiology of invasive infections due to non-betahemolytic streptococci, France, 2021" by Céline Plainvert et al., describes the clinical and microbiological epidemiology of invasive infections due to non-beta-hemolytic streptococci during the period March and April 2021, along with the screening of their behavior towards different antimicrobial agents. The paper reported a relevant issue, however many weaknesses are highlighted into the text.

Major points to address:

Question 1) Even if the authors sampled 522 invasive NBHS between March and April 2021; I really believe that it is too dated since, until now, additional period should have been considered to strengthen the results. In the manuscript the authors identified and characterized bacteria of 2 years ago, considering the speed at which bacteria acquire resistance, the picture they report is not current at all. This is a crucial limitation of the study.

- We agree with the reviewer about the fact that the isolates analyzed in this study are a bit dated and that adding strains isolated last year would probably better reflect the current epidemiology of resistance to antibiotics in the corresponding species. However, we would like to bring a few points to the attention of the reviewer :

- first of all, although the epidemiology of resistance is rapidly evolving, we believe that it does not change fundamentally from year to year, and adding isolates from last year, i.e. only one year after our study was performed, would hardly change the conclusions of the paper; in their study including over 4,800 *viridans* group streptococci over a 10-year study-period, Singh et al. found that resistance rates were stable (JAC-Antimicrobial resistance 2022).

- secondly, the results presented are those of an *ad hoc* survey carried out among our correspondents in order to obtain all the strains responsible for invasive infections, avoiding the usual recruitment biases that lead our correspondents to send us only the most resistant strains or those associated with the most severe infections; to correctly update the data, it would be necessary to repeat the same type of study, which represents a significant additional workload and an important delay for the publication of our results; performing the experiments and analyzing the results (which included 522 BMD plates plus a significant number of repeats to consolidate the results together with the gradient tests, disk diffusion and the EUCAST laboratory) represents a significant amount of minimum 6 months of work;

- last, the study was designed to provide a snapshot of the epidemiology of invasive infections due to non-beta-hemolytic streptococci which are largely overlooked and to serve as a baseline for subsequent studies which we actually plan to perform every 3 to 5 years.

Nevertheless, we understand the reviewer's concern and we analyzed the results of non-beta-hemolytic streptococcal isolates responsible for invasive infections sent to the national reference center in 2022. As specified above, there was no specific request for these isolates in 2022 and they only represented a total of 79 isolates out of the 2,292 total isolates received that year (mainly group A and B streptococci). Given that the 79 isolates were mainly sent for confirmation of their identification or antibiotic resistance phenotype and that they are largely outnumbered by the isolates of March-April 2021, we did not include them to our previous results but showed their analysis in a distinct paragraph, lines 214-232. The results are also provided as supplemental material (Table S7 to S10). Of note, susceptibility testing of these

79 isolates was performed by gradient test for beta-lactams and by disk diffusion for the other molecules. Overall, because there is a significant difference in the total number of strains analyzed in each time-period and because of the recruitment biases in the strains collected in 2022, it is difficult to compare the two sets of results. Still, the main conclusions are as follows:

- clinical features: the distribution among the different streptococcal groups and between adult and pediatric cases seems similar;
- susceptibility to beta-lactam agents: all isolates of the *S. anginosus* and *S. bovis/equinus* groups remain susceptible to beta-lactam agents; resistance rates of the *S. mitis* group seem to have increased in 2022 (approx. 60% vs. 30% in 2021) but one must remember that most of these isolates were sent for confirmation of their resistance phenotype making it difficult to interpret these data; resistance rates of the *S. salivarius* and *S. sanguinis* groups are not interpretable because of the small number of isolates (4 and 7, respectively);
- susceptibility to macrolides, lincosamides, streptogramins, and tetracyclines: the resistance rates, the phenotypes, and the distribution of resistance determinants are similar in 2022 to those in 2021.

Beside presenting these results, we discussed them and the related biases in the discussion section, lines 236-239, 246-249, and 329-332.

2) The introduction section is too generic. The authors did not did not deepen the numerical relevance of the impact of the following sentences: "These commensals are responsible for a wide range of invasive diseases, particularly in the presence of underlying conditions and in immune-compromised adults. Importantly, NBHS are reported as the first or second cause of infective endocarditis worldwide [3-5]. Additionally, isolates of the *Streptococcus anginosus* group are causative agents of pyogenic infections and abscesses in immunocompetent adults and children [6]."

- We understand the reviewer's comment and specified more accurately the importance of NBHS infections in the first paragraph of the introduction, lines 67-85.

3) In the discussion, the authors present only few comparisons with literature data; the same concepts are repeated through the text without creating a logical and continuous argumentation. The last paragraph of the manuscript in addition to the overall conclusion of the research should contain the limitations of the study. The author should rewrite the discussion to increase its overall quality.

- We apologize if the discussion did not seem of good quality to the reviewer. We tried to make it clear and concise while complying with the guidelines recommended by Docherty and Smith: BMJ 1999;318:1224-5; namely: summary of the principal findings; findings of the present study in light of what was published before; strengths and limitations of the study; meaning of the study; understanding possible mechanism; implications for practice or policy; implications for future research.

We took note of the reviewer's comment and rewrote the discussion to deepen comparisons with literature data and we recalled the main limitations of the study in the last paragraph as requested.

In the previous version of the manuscript, comparison with literature data (which are scarce) were presented lines 204-215. We added more elements in the revised version of the manuscript lines 258-286.

Limitations of the study were discussed lines 232-238. We recalled the main one in the last paragraph of the revised version line 352.

Minor points

Please check all the text making attention in abbreviations (for that reason it should be in extensor the first time mentioned and then the abbreviation) and in bacterial nomenclature (i.e. prefer the term *S. mitis* instead of *mitis*, and write *Streptococcus* spp. when appropriate).

- We apologize for the misspelling and corrected the text accordingly.

April 29, 2023

Dr. Asmaa Tazi
Assistance Publique - Hopitaux de Paris
Paris
France

Re: Spectrum00160-23R1 (Microbiological epidemiology of invasive infections due to non-beta-hemolytic streptococci, France, 2021)

Dear Dr. Asmaa Tazi:

Your manuscript has been accepted, and I am forwarding it to the ASM Journals Department for publication. You will be notified when your proofs are ready to be viewed. Based on the reviewers' suggestion, please have an experienced native English speaker review and edit the revised discussion section to ensure clarity, coherence, and proper use of terminology. This will make the text more accessible and easier to understand for readers, maximizing the impact of the study.

Sincerely,

Po-Yu Liu
Editor, Microbiology Spectrum
